# Factors associated with contraceptive use among reproductive-age women during a pandemic: Evidence from a small developing state

Clemon George[1]*, Heather Harewood[2], Michael Campbell[2], Keerti Singh[2], Eden Hope Augustus[3]

**1** Health, Hospitality, Nutrition and Dietetics Department, School of Education and Applied Professions, Buffalo State University, Buffalo, New York, United States of America, **2** Faculty of Medical Sciences, University of the West Indies, Bridgetown, Barbados, **3** The George Alleyne Chronic Disease Research Institute, University of the West Indies, Bridgetown, Barbados

* georgec@buffalostate.edu

## Abstract

The COVID-19 pandemic disrupted access to sexual and reproductive health services, including contraception, highlighting the need to understand contraceptive use to strengthen health-system resilience in small developing states. This study examined factors associated with contraceptive use among reproductive-age women during the early phase of the pandemic in Barbados. A cross-sectional online survey was conducted in 2020 among adults aged 18 years and older, with analysis restricted to non-pregnant women aged 18–49 years. Contraceptive use was assessed in relation to sociodemographic characteristics, relationship status, and psychosocial distress measured using the Hospital Anxiety and Depression Scale. Multiple imputation was used to address missing data, and sensitivity analyses compared imputed and complete-case models. Among 1,094 women, 34.2% (n = 333) reported current contraceptive use, while 36.5% reported moderate-to-severe anxiety and 39.0% reported clinically significant distress. In multivariable analyses, tertiary education (OR 1.68; 95% CI 1.24 - 2.28), being partnered but not cohabiting (OR 1.57; 95% CI 1.10 - 2.22), and being married or cohabiting (OR 2.49; 95% CI 1.65 - 3.77) were associated with higher odds of contraceptive use. Anxiety symptoms showed a weak positive association, while increasing age was associated with slightly lower odds. Findings were consistent across sensitivity analyses. These results underscore the importance of maintaining access to sexual and reproductive health services and integrating mental health support into emergency preparedness strategies.

## Introduction

Sexual and reproductive health (SRH) is a core component of global health strategies and is grounded in the recognition of sexual health as a fundamental human right,

**Data availability statement:** The raw data underlying this study have been deposited in a publicly accessible repository on Figshare and are available at https://doi.org/10.6084/m9.figshare.32099512.

**Funding:** The authors received no specific funding for this work.

**Competing interests:** The authors have declared that no competing interests exist.

including the right to access family planning services and comprehensive reproductive care [1]. Despite this recognition, SRH services are frequently deprioritized during public health emergencies. During the early phase of the COVID-19 pandemic, disruptions to health systems worldwide substantially reduced access to SRH services, with women disproportionately affected [2,3].

Although a growing body of literature has documented the impact of COVID-19 on SRH services, particularly in low- and middle-income countries, these data often fail to capture the lived realities of small island developing states (SIDS), including those in the Caribbean [3]. Because of their small populations, limited health infrastructure, geographic isolation, and constrained fiscal space, Caribbean SIDS experience distinct vulnerabilities that shape access to and quality of healthcare services [4,5]. Regional analyses frequently aggregate Caribbean countries with Latin America, masking heterogeneity and obscuring the specific needs of smaller island nations. In these settings, health systems are shaped by different structural constraints, including limited economies of scale due to small population sizes, persistent health workforce shortages, geographic isolation, dependence on imported medical supplies, and heightened vulnerability to economic instability and climate-related disasters [6].

Over the past three decades, Caribbean countries have made notable progress in advancing sexual and reproductive rights following the articulation of SRH as a global health priority [7]. Legislative and policy reforms have strengthened protections for women's and children's health. For example, earlier legislative and policy environments in several Caribbean countries limited adolescents' access to sexual and reproductive health (SRH) services and, in some cases, resulted in the exclusion of pregnant adolescents from formal education. Recent reforms have sought to address these gaps through the introduction of adolescent-responsive SRH policies, including the application of competency-based guidelines (e.g., Gillick competence), the implementation of school re-entry policies for pregnant adolescents, and the expansion of legal protections against gender-based violence. Collectively, these reforms have strengthened access to essential services and enhanced protections for women and children across the region however, implementation gaps persist [7]. These gaps are driven by limited resources, workforce constraints, and uneven translation of policy into service delivery [6,8]. Moreover, the region's exposure to natural disasters, including hurricanes and other climate-related shocks, regularly disrupts economic and social systems, further threatening the continuity of SRH services [7].

In recognition of crisis-related vulnerabilities, many Caribbean countries have invested in preparedness frameworks such as the Minimum Initial Service Package (MISP) for SRH in emergencies, now considered a global standard [7]. While progress has been made in policy adoption, important gaps remain in research capacity, sustainable financing, and operational implementation. For example, although several countries report high levels of domestic financing for HIV and reproductive health programs, data on funding and service provision including gender-based violence, broadly encompassing intimate partner, domestic, and sexual violence, prevention and response are limited [9]. Furthermore, monitoring and evaluation of

SRH programs in the region often rely on administrative reporting or anecdotal evidence rather than systematic empirical research [9,10].

Contraceptive availability and use remain important to improving SRH outcomes, as they directly influence women's reproductive autonomy, economic participation, and overall wellbeing [11]. According to recent regional assessments by United Nations Population Fund, progress toward meeting contraceptive needs in the Caribbean has been uneven [7]. Contraceptive methods remain heavily skewed toward male condoms and injectables, while access to long-acting reversible contraceptives and emergency contraception remains limited in many countries [12,13]. These structural limitations increase vulnerability to service disruptions during crises.

Globally, the consequences of pandemic-related SRH disruptions have been substantial. In 2021, UNFPA estimated that approximately 12 million women experienced interruptions in contraceptive services, resulting in 1.4 million unintended pregnancies, the majority occurring in low- and middle-income countries [14]. Small developing states are particularly susceptible to such disruptions due to limited-service redundancy and fragile supply chains. In the Caribbean, these vulnerabilities are reflected in persistently high adolescent birth rates, which remain among the highest worldwide [15]. In 2023, the adolescent birth rate in Latin America and the Caribbean was estimated at 51.4 births per 1,000 girls aged 15–19, with many pregnancies unintended and associated with increased risk of unsafe abortion and adverse maternal outcomes.

Despite these concerns, empirical evidence on the impact of COVID-19 on SRH in Caribbean SIDS remains sparse. A major scoping review of 83 studies on SRH service disruptions during the pandemic included only four studies from Latin America and the Caribbean, highlighting a significant regional evidence gap [2]. Available studies indicate widespread reductions in access to contraception, abortion care, STI/HIV services, and gender-based violence support, even in settings with relatively well-resourced health systems that adopted telehealth models [16]. Adolescents and other vulnerable populations appear to have been disproportionately affected due to school closures, reduced youth-friendly services, and prolonged periods of unstructured time [2].

In the Caribbean context, limited data infrastructure and variation in pandemic mitigation strategies complicate efforts to systematically assess COVID-19's impact on SRH. At the same time, close-knit social networks, household structures, and gendered economic roles may shape reproductive decision-making in ways that differ from larger or more resourced regions [17].

This study investigates the factors associated with contraceptive use during the pandemic, with a focus on psychosocial health, education, age, and relationship status in a small island developing country in the Caribbean.

## Methods

### Ethics statement

This study presented minimal risk to participants. The survey consisted of self-reported questions related to experiences during the COVID-19 pandemic and did not involve interventions or procedures expected to cause harm. The level of risk was considered no greater than that encountered in routine daily life during the pandemic. Participation was voluntary, and safeguards were implemented to ensure confidentiality and data protection. No incentive was offered to participants. Upon completion, participants were provided with referral information for optional governmental community-based social support resources. IRB approval was obtained from the University of the West Indies – Cave Hill/Barbados Ministry of Health IRB No.200405-B.

### Study design and participants

This study is part of a larger cross-sectional survey of Barbadian experiences of COVID-19 and associated public health control measures during the early phases of the pandemic. The survey instrument utilized a modified version of the *Public Response to UK Government Recommendations on COVID-19: Population Survey, 17–18 March 2020* questionnaire [18].

The survey link was shared as text blasts to subscribers by two major Barbadian cellphone service providers alongside their product marketing. Additionally, recruitment continued by snowballing via personal, religious, media, and other civil society networks. In Barbados, these two dominant cell phone providers often advertise products to their clients through text blasts with extensive reach. The number of cell phone connections in 2020 was equivalent to 117% of the population of Barbados [19].

On opening the survey link, participants needed to read informed consent information and give approval before completing the survey. Eligible participants ≥18 years and resident in Barbados for ≥3 months, were recruited between April 28 and May 3, 2020. The participant information sheet detailed the purpose of the survey, possible risks and benefits, data protection information, and the right to withdraw from the survey without penalty. It was estimated that the survey would take approximately 25 minutes to complete.

The survey included questions on participants' knowledge of COVID-19, current and perceived impact of the pandemic on people's health and well-being. Additionally, questions on pregnancy, use of contraceptives, and access to contraceptives were included, as were items related to sociodemographic, general health, and mental health status, including the Hospital Anxiety and Depression Scale (HADS) [20].

## Measures

The sample for this analysis includes only adult women of reproductive age as defined by the World Health Organization (we used 18–49 as in Barbados, those under 15 years are considered children). Current contraceptive use (yes/no) was assessed among women who reported not being pregnant at the time of the survey. Pregnancy status was measured first; women who were pregnant were excluded from contraceptive-use analyses. Contraceptive use was measured using the question: "Have you been using any method of birth control over the past two months?" Reasons for non-use were assessed with a multi-option item allowing respondents to select applicable reasons (I did not expect to have sex, I am worried about the side effects of contraception, my male partner does not want me to use contraception, etc.). Independent variables included: Mental health status assessed through the Hospital Anxiety and Depression Scale (HADS) (Anxiety (HADS-A), Depression (HADS-D), and global score; relationship status (single, in a relationship, married, divorced/separated); education (tertiary education (university degree) versus lower); age and other behavioral variables such as marijuana and alcohol use three months before the pandemic. Alcohol use was measured as daily, a few times weekly (frequently), a few times monthly (occasionally), less than once a month (rarely) and never (or having stopped more than 3 months prior to the pandemic).

Statistical Analysis: Univariable logistic regression analyses were conducted on complete-case data as exploratory assessments of crude associations, while primary inference was based on multivariable models fitted to multiply imputed datasets to address missing data and estimate adjusted effect sizes.

Descriptive statistics were generated for key study variables. Associations between current contraceptive use (yes/no) and each independent variable were examined using bivariate methods. For categorical predictors, we estimated univariable logistic regression models (one predictor at a time) and evaluated statistical significance using likelihood ratio (LR) tests. As some variables showed up to 15% missing, the second step was to address this using multiple imputation by chained equations (MICE) under the assumption that the data were missing at random (MAR). These included alcohol and marijuana use, relationship status, education, contraceptive use and items of the Hospital Anxiety and Depression Scale (HADS), were also imputed using conditional models based on their measurement scale. Twenty imputed datasets were generated following a burn in period, and estimates were combined using Rubin's rules [21]. HADS anxiety and depression scores were calculated after imputation by summing the relevant responses, ensuring that scale scores were derived from imputed item-level data. Continuous HADS scores were used in regression analyses to preserve statistical power. Age was included as a continuous variable in the regression models because it contained no missing data. Model diagnostics, including the relative variance increase and fraction of missing information, were examined to assess the

impact of missing data. Multivariable logistic regression was then performed across the imputed data sets to estimate associations with contraceptive use. Statistical significance was assessed using two-sided tests with a significance level of 0.05. Univariate and bivariate analyses were first conducted in SPSS v28. MICE and multivariable logistic regression were conducted in Stata.

## Results

A total of 1,094 women aged 18–49 years completed the survey and were included in the analyses (Table 1). At the time of the survey, 2.7% (n = 29) reported being pregnant and 7.3% (n = 80) reported not being pregnant but planning pregnancy; the majority (89.7%, n = 981) were neither pregnant nor planning pregnancy.

Among non-pregnant women with contraceptive data, 34.2% (n = 333) reported current contraceptive use and 65.8% (n = 640) reported no contraceptive use. Among contraceptive users reporting method type, short-acting reversible methods were most common (53.7%, n = 194), followed by long-acting reversible contraception (24.4%, n = 88). Permanent (3.6%), natural methods (2.2%), and other methods were less frequently reported.

### Psychosocial distress and contraceptive use

Symptoms of anxiety and depression were common during the early phase of the pandemic. Based on HADS subscales, 36.5% of respondents reported moderate-to-severe anxiety symptoms (levels 2–3), and 41.0% reported moderate-to-severe depressive symptoms (levels 2–3). For the HADS global score, 39.0% of women reported moderate or clinically significant distress, including 14.1% classified as clinically significant.

Contraceptive use was significantly associated with anxiety symptoms and overall distress. Severe HADS anxiety scores were associated with higher odds of contraceptive use (OR 1.51; 95% CI 1.05-2.16), indicating greater likelihood of use compared with the reference group. Similarly, clinically significant HADS global distress was significantly associated with contraceptive use (OR 1.54; 95% CI 1.04-2.29). In contrast, HADS depressive symptoms alone were not significantly associated with contraceptive use.

### Sociodemographic and relationship correlates

Relationship status was strongly associated with contraceptive use. Compared with single women, those who were partnered but not cohabiting were more likely to use contraception (OR 1.54, 95% CI 1.10-2.16), and those married or cohabiting showed a stronger association (OR 2.52; 95% CI 1.80-3.52). No association was observed among women who were separated, divorced, or widowed.

Education also showed a strong association: women with university education were significantly more likely to report contraceptive use than those with lower educational attainment (OR 1.64; 95% CI 1.24 - 2.17).

Age was associated with contraceptive use, with women aged 26–39 having a statistically significant association with contraceptive use compared to women aged 40–49 years, while women aged 18–25 years having no statistical significance.

### Household context and health behaviors

Nearly half of respondents reported being the main household income earner (46.6%), while approximately one-third reported that parents/grandparents were the primary earners (33.2%). In breadwinner comparisons, having a partner as the primary income earner was significantly associated with contraceptive use relative to self-earners (OR = 2.04; 95% CI 1.30 - 3.20).

Most women (84.6%) reported not stockpiling condoms or birth control during the early pandemic. Alcohol and marijuana use were examined but had >15% missing data; nonetheless, marijuana but not alcohol demonstrated positive bivariate associations with contraceptive use (Table 2). These findings should be interpreted with caution.

**Table 1. Sociodemographic, health, and psychosocial characteristics of reproductive-age women in Barbados, 2020.**

| Characteristic | n | % |
|---|---|---|
| Age group (years) | | |
| 18–25 | 237 | 21.6 |
| 26–39 | 549 | 50.2 |
| 40–49 | 308 | 28.2 |
| Relationship status | | |
| Single | 364 | 33.8 |
| Partnered (not cohabitating) | 348 | 32.3 |
| Married/cohabitating | 333 | 30.9 |
| Separated/divorced/widowed | 31 | 2.9 |
| Education | | |
| Less than high school | 18 | 1.7 |
| High school | 159 | 15.2 |
| College | 247 | 23.6 |
| University | 621 | 59.4 |
| Race/ethnicity | | |
| Black | 873 | 83.3 |
| East Asian | 4 | 0.4 |
| White | 23 | 2.2 |
| Indian | 68 | 6.5 |
| Mixed | 80 | 7.6 |
| Children in household | | |
| None | 530 | 49.4 |
| One | 269 | 25.1 |
| Two | 179 | 16.7 |
| Three | 61 | 5.7 |
| Four or more | 33 | 3.1 |
| General health status | | |
| Good | 288 | 27.2 |
| Fairly good | 519 | 49.0 |
| Average | 189 | 17.8 |
| Poor | 63 | 5.9 |
| Mental health (HADS) | | |
| Anxiety – none/mild | 631 | 63.5 |
| Anxiety – moderate/severe | 363 | 36.5 |
| Depression – none/mild | 600 | 59.1 |
| Depression – moderate/severe | 415 | 40.9 |
| Global distress – none/mild | 646 | 61.0 |
| Global distress – moderate/clinically significant | 409 | 39.0 |
| Current contraceptive use* | | |
| Yes | 333 | 34.2 |
| No | 640 | 65.8 |

Note: [a.]Percentages may not sum to 100 due to rounding.

[b.]Contraceptive use assessed among non-pregnant women only.

**Table 2. Univariable associations between selected factors and current contraceptive use** *(Likelihood ratio tests from single-predictor logistic regression models).*

| Variable | Comparison/ Reference | OR | 95% CI | p-value |
|---|---|---|---|---|
| HADS Anxiety | None/mild (ref) | | | |
| | Moderate | 1.22 | 0.85-1.76 | 0.288 |
| | Severe | **1.51** | **1.05-2.16** | **0.026** |
| HADS Depression | None/mild (ref) | | | |
| | Moderate | 0.90 | 0.64-1.27 | 0.559 |
| | Severe | 1.34 | 0.94-1.94 | 0.106 |
| HADS Global Distress | None/mild (ref) | | | |
| | Moderate | 1.30 | 0.94-1.78 | 0.109 |
| | Clinically significant | **1.54** | **1.04-2.29** | **0.033** |
| Relationship status | Single (ref) | | | |
| | Partnered (not cohabitating) | **1.54** | **1.10-2.16** | **0.014** |
| | Married/cohabitating | **2.52** | **1.80-3.52** | **<0.001** |
| | Separated/divorced/widowed | 0.75 | 0.29-1.83 | 0.490 |
| Education | University vs. non-university | **1.64** | **1.24-2.17** | **<0.001** |
| Age group (years) | 40–49 (ref) | | | |
| | 33–39 | **2.47** | **1.69-3.61** | **<0.001** |
| | 26–32 | **2.12** | **1.46-3.08** | **<0.001** |
| | 18–25 | 1.23 | 0.83-1.81 | 0.297 |
| Alcohol use (pre-COVID)† | Never (ref) | | | |
| | Rarely | 1.07 | 0.42-2.69 | 0.89 |
| | Occasionally | 1.30 | 0.52-3.20 | 0.58 |
| | Frequently | 1.11 | 0.43-2.89 | 0.82 |
| | Daily | 2.23 | 0.86-5.78 | 0.10 |
| Marijuana use | Never vs. ever | 1.59 | 1.03-2.49 | 0.036 |

Note: [a] LR = likelihood ratio test from univariable logistic regression (one predictor at a time).

[b] Bold indicates p < 0.05.

[c] Marijuana and Alcohol had > 15% missing data. Missing values are imputed in the MLR (Table 3).

[d†] Never = Never or stopped using alcohol > 3 prior to the pandemic; Rarely = alcohol use less than monthly; Occasionally = alcohol use less than weekly; Frequently = weekly alcohol use; Daily = daily alcohol use.

## Reasons for non-use of contraception

Among women who were not pregnant and not using contraception, common reasons included not expecting to have sex (17.1%), concerns about side effects (13.2%), and believing pregnancy was unlikely (9.3%). Partner opposition to contraception or condoms was infrequently endorsed (<1%). A sizable proportion selected "other" (20.8%), suggesting heterogeneity in barriers and motivations.

## Multivariable analysis

Logistic regression model estimates using 20 multiple imputed datasets indicated that contraceptive use was significantly associated with education, relationship status and age. Table 3 shows that individuals with tertiary education had higher odds or contraceptive use compared to those without tertiary education (OR 1.68; 95% CI: 1.24 - 2.28). Compared to single individuals, those in partnerships but non-cohabiting (OR 1.57; 95% CI 1.10 - 2.22) and those who were married or cohabitated (OR 2.49, 95% CI 1.65 - 3.77) had significantly greater odds of contraceptive use. In bivariate analyses,

**Table 3. Multivariable associations between selected factors and current contraceptive use based on MICE.**

| Variable | Odds Ratio | 95% CI |
| --- | --- | --- |
| HADS Anxiety score (continuous) | 1.06 | 1.00–1.12 |
| HADS Depression score (continuous) | 0.98 | 0.92–1.05 |
| Partner not cohabiting | 1.57 | 1.10–2.22 |
| Married/cohabiting | 2.49 | 1.65–3.77 |
| Separated/divorced/widowed | 0.90 | 0.34–2.35 |
| Tertiary education | 1.68 | 1.24–2.28 |
| Age (per year) | 0.98 | 0.95–1.00 |
| Alcohol use (level 2) | 1.22 | 0.85–1.76 |
| Alcohol use (level 3) | 0.97 | 0.61–1.54 |
| Alcohol use (level 4) | 1.53 | 0.95–2.46 |
| Marijuana use | 1.27 | 0.80–2.03 |

women aged 26–32 and 33–39 differed significantly in contraceptive use compared to those aged 40–49, while no significant difference was observed for women aged 18–25. In multivariable analysis treating age as a continuous variable, increasing age was associated with a modest reduction in contraceptive use (OR = 0.98 per year, 95% CI 0.95 - 1.00). Anxiety symptoms showed a marginal positive association with contraceptive use (OR 1.06, per unit increase, p = 0.066), whereas depressive symptoms, marijuana consumption and breadwinner status were not significantly associated with contraceptive use. Figure 1 shows a forest-plot of the adjusted odds ratios.

Sensitivity analyses comparing the imputed model with a complete-case logistic regression showed similar results. Differences between models were primarily reflected in narrower confidence intervals in the complete-case analysis (see S1 Table).

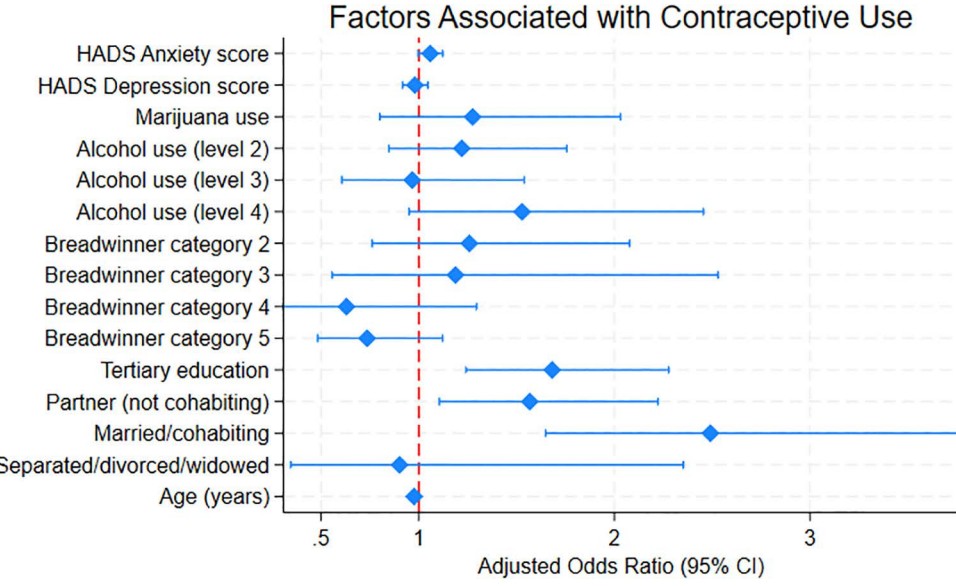

**Fig 1. Forest plot: adjusted odds ratios for factors associated with contraceptive use from multiple imputed logistic regression models.** [Forest plot will be uploaded separately as it reformats in Word®.].

## Discussion

This study examined factors associated with contraceptive use among reproductive-age women during the early phase of the COVID-19 pandemic in a Caribbean small island developing state. Consistent with global evidence of widespread disruption to sexual and reproductive health (SRH) services during the pandemic, contraceptive use in this sample was patterned by relationship status, education, and age. These findings reflect observed statistical associations and should be interpreted as descriptive rather than causal, consistent with the cross-sectional design.

Anxiety symptoms were modestly and positively associated with contraceptive use in both univariable and multivariable analyses.

In multivariable analysis, anxiety symptoms showed a modest positive association with contraceptive use, whereas depressive symptoms were not associated. This pattern aligns with literature indicating that anxiety, especially during periods of acute uncertainty, may heighten risk-avoidant or protective health behaviors, including efforts to avoid unintended pregnancy [22]. During the early COVID-19 period, anxiety related to economic instability, health system strain, and uncertainty about access to care may have contributed to increased contraceptive vigilance among some women [23].

In contrast, depressive symptoms were not associated with contraceptive use in the adjusted model, consistent with literature linking depression to reduced health-seeking behavior and diminished agency [24,25]. The drop of associations after adjustment suggests that some of the stronger relationships observed in unadjusted analyses may have been influenced by confounding factors.

The distinction between anxiety and depression is rarely examined in sexual and reproductive health research during emergencies, particularly in small developing states. These findings underscore the importance of disaggregating psychosocial measures rather than relying on composite mental health indicators alone.

For Caribbean SIDS where mental health services are often under-resourced and stigmatized [26], the observed pattern suggests that psychological factors, particularly anxiety, may still play a role in shaping contraceptive behavior, highlighting the potential value of integrating mental health screening and support into SRH service delivery during crises.

Consistent with prior research [27,28], women who were partnered or married were more likely to report contraceptive use compared with single women. This finding reflects well-documented patterns in which contraceptive use is shaped by relationship context, perceived pregnancy risk, and negotiated decision-making within partnerships [29,30]. In the Caribbean context, where fertility norms, union stability, and gendered expectations around contraception vary widely [31,32], relationship status may serve as a proxy for both sexual activity patterns and access to partner support for contraceptive use.

Educational attainment also showed a strong association with contraceptive use, with women holding university degrees more likely to report use than those with lower education. This finding aligns with global evidence linking education to health literacy, reproductive autonomy, and navigation of health systems [33,34]. In small island settings, where method choice may already be constrained by supply limitations, education may further influence awareness of available options and confidence in engaging with providers.

Age differences in contraceptive use were evident, with women aged 26–39 years more likely to report use than those aged 40–49 years, while women aged 18–25 years were less likely. These patterns are consistent with the multivariable analysis, which showed a modest inverse association between age and contraceptive use when modeled continuously, suggesting a gradual decline in use with increasing age. Lower use among younger women aligns with global and regional evidence of persistent unmet contraceptive need among adolescents and young adults [35,36]. During the pandemic, school closures, reduced youth-friendly health services, and increased reliance on household or parental environments may have further constrained access for younger women in Caribbean settings. This finding is particularly salient given persistently high adolescent and young adult pregnancy rates in the region [37,38]. It suggests that pandemic-related disruptions may have reinforced pre-existing barriers to contraceptive access among younger women rather than creating entirely new ones.

Although this study did not directly measure service availability, the low prevalence of contraceptive stockpiling reported by participants suggests limited anticipatory behavior in response to potential access disruptions. In SIDS, where contraceptive supply chains are highly centralized and dependent on external procurement, even short interruptions can disproportionately affect access [4,39]. However, in our case, the population may have been conditioned by a relative ease of access under normal circumstances, which points to some measured success of Maternal and Child Health program [40]. UNFPA reported challenges in supply chain management, the logistics management information system and the inventory control systems in reproductive health commodity security across Caribbean countries with regards to oversupply in some areas and undersupply in others, across the different countries [5]. The predominance of short-acting methods observed in this sample further underscores system-level vulnerability, as these methods require regular resupply and consistent service contact.

These findings are consistent with regional reports indicating uneven contraceptive method mix and limited availability of long-acting reversible contraception across Caribbean countries [12,13]. During emergencies, such structural constraints may amplify the impact of health system shocks on reproductive autonomy.

## Contribution to the literature and regional relevance

This study contributes new empirical evidence from a Caribbean SIDS, addressing a notable gap in the global literature on SRH during COVID-19. While numerous studies have documented SRH disruptions in larger low- and middle-income countries, Caribbean SIDS remain under-represented despite their unique structural vulnerabilities. By linking psychosocial distress, sociodemographic characteristics, and contraceptive use during the early pandemic phase, this analysis adds context-specific insight that is often obscured in regional aggregates.

The findings are particularly relevant for emergency preparedness planning in SIDS, where limited service redundancy, small workforces, and fragile supply chains increase the importance of anticipatory, integrated SRH strategies. The observed associations underscore the need to maintain contraceptive access alongside mental health support, especially for younger women and those with lower educational attainment.

## Limitations

Several limitations should be acknowledged. As a cross-sectional study design, temporal relationships between age, psychological factors, education, relationship status, and contraceptive use cannot be established, limiting causal inference. Although multivariable modeling was used to adjust for measured confounders, residual confounding from unmeasured or imprecisely measured variables including fertility intentions, sexual activity, or access barriers remains possible. Some covariates contained missing data and were addressed using multiple imputation by chained equations under a missing-at-random assumption prior to multivariable logistic regression. While this approach preserves sample size and is generally preferable to complete-case analysis, the validity of the estimates depends on the adequacy of the imputation model and the plausibility of the missing-at-random assumption; therefore, some bias may persist if missingness was not fully explained by observed variables.

In addition, key variables were modeled in ways that may not fully capture their underlying complexity. Age was examined both categorically and continuously, and while broadly consistent patterns were observed, the continuous specification assumes a linear relationship that may obscure non-linear differences across reproductive age groups. Psychological distress was assessed using separate anxiety and depression scores, and although anxiety showed a modest, borderline association with contraceptive use while depression did not, these relationships may be sensitive to model specification and residual confounding. Similarly, education and relationship status were included as categorical variables, but these may not fully reflect underlying socioeconomic position, partnership dynamics, or access to resources that influence contraceptive behavior. These findings should therefore be interpreted with caution in light of both the cross-sectional design and the imputation process. Finally, data were collected during the early phase of the pandemic and may not capture subsequent adaptations in service delivery.

## Implications for policy and future research

Despite these limitations, the findings highlight actionable priorities for SRH policy in small developing states, especially in times of continual funding and policy changes of large donors [41]. Emergency response planning should prioritize continuity of contraceptive services, expand access to self-managed and long-acting methods where feasible, and integrate mental health considerations into SRH programming. Future research should employ longitudinal designs and multivariable analyses to better understand pathways linking psychosocial distress and reproductive decision-making during crises, particularly among adolescents and young adults in SIDS contexts.

## Supporting information

**S1 Table. Sensitivity analysis.**
(DOCX)

## Author contributions

**Conceptualization:** Clemon George, Heather Harewood, Michael Campbell, Keerti Singh, Eden Hope Augustus.

**Data curation:** Clemon George.

**Formal analysis:** Clemon George.

**Investigation:** Clemon George, Heather Harewood, Eden Hope Augustus.

**Methodology:** Clemon George, Heather Harewood, Michael Campbell, Eden Hope Augustus.

**Project administration:** Clemon George.

**Software:** Clemon George.

**Validation:** Clemon George, Keerti Singh.

**Visualization:** Clemon George, Michael Campbell.

**Writing – original draft:** Clemon George.

**Writing – review & editing:** Clemon George, Heather Harewood, Michael Campbell, Keerti Singh, Eden Hope Augustus.

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
