## [Decision Letter · Decision Letter 0]

2 Mar 2026

PGPH-D-26-00542

Factors associated with contraceptive use among reproductive-age women during a pandemic: Evidence from a small developing state

Dear Dr. George,

Thank you for submitting your manuscript to PLOS Global Public Health. After careful consideration, we feel that it has merit but does not fully meet PLOS Global Public Health’s publication criteria as it currently stands. Therefore, we invite you to submit a revised version of the manuscript that addresses the points raised during the review process.

EDITOR:Please carefully review and respond to all comments from the reviewers. Note that some comments overlap between Reviewer 1 and Reviewer 2, so you may address these duplicate points together. Prepare your responses in a separate document using a point-by-point format for each reviewer’s comments. In the revised manuscript, please indicate all changes using tracked changes and/or highlighting them.

We look forward to receiving your revised manuscript.

Kind regards,

Tanmay Bagade, Ph.D., MS (O&G), MPH, MHM

Academic Editor

Journal Requirements:

1. We have noticed that you have uploaded Supporting Information files, but you have not included a list of legends. Please add a full list of legends for your Supporting Information files after the references list.

2. In the online submission form, you indicated that "The data that support the findings of this study are available from the corresponding author upon reasonable request.".

3. Uploaded as supplementary information.

Additional Editor Comments (if provided):

Reviewers' comments:

Reviewer's Responses to Questions

**Comments to the Author**

1. Does this manuscript meet PLOS Global Public Health’s publication criteria? Is the manuscript technically sound, and do the data support the conclusions? The manuscript must describe methodologically and ethically rigorous research with conclusions that are appropriately drawn based on the data presented.

Reviewer #1: Yes

Reviewer #2: Yes

Reviewer #3: Yes

Reviewer #4: Yes

2. Has the statistical analysis been performed appropriately and rigorously?

Reviewer #1: No

Reviewer #2: No

Reviewer #3: No

Reviewer #4: Yes

3. Have the authors made all data underlying the findings in their manuscript fully available (please refer to the Data Availability Statement at the start of the manuscript PDF file)?

Reviewer #1: Yes

Reviewer #2: Yes

Reviewer #3: No

Reviewer #4: No

4. Is the manuscript presented in an intelligible fashion and written in standard English?

Reviewer #1: Yes

Reviewer #2: Yes

Reviewer #3: Yes

Reviewer #4: Yes

5. Review Comments to the Author

Reviewer #1: First of all, I would like to thank all the authors for conducting research on such sensitive and often neglected issues, especially at the time of health emergencies. You have done such good work. Here are some of my reviews to help strengthen and support your study. These are my views and you have every right to disagree with it.

Abstract

Results: The abstract says that logistic analysis was performed but only p-values are reported. For logistic regression, reporting odds ratios with 95% Confidence Intervals is necessary to interpret the magnitude and direction of association. Again, Variables, alcohol and marijuana use had >15% missing data which limited? Limited what? The language is very unclear and this whole makes it confusing for the readers to understand.

Introduction

Line 39,40,41: “During the early phase of the COVID-19 pandemic, disruptions to health systems worldwide substantially reduced access to SRH services, with women disproportionately affected….”

Comment: The authors have used the 4 references for a sentence that seems a bit overcrowded. Do this all articles say the same? Or the other were proving the point of their sentence than providing the reference?

Line 50: “systems operate under markedly different structural constraints” What kind of structural constraints? It would have been more informative If the types of structural constraints are mentioned

Line 53,54: “Legislative and policy reforms have strengthened protections for women’s and children’s health; however, implementation gaps persist”. What were the previous legislative policies, and what are the new ones that have helped in strengthening the protection for women’s and children’s health? I think an example would provide a context and help readers grasp the underlying issues.

Line 64,65: “data on funding and service provision for gender-based violence prevention and response are limited”

I think this sentence needs a bit more clarity, authors have mentioned about the provision for GBV but does this GBV explicitly refer to SRH or other domestic issues?

Methods

Line 146 to154: Same repetition of the question for assessing the contraceptive use, again the repetition of age-group.

Line 182 the table description can be kept as in a single column such as; n(%) 237(21.6%).

Line 184: “Percentages may not add up to 100 due to rounding…. I think this can be kept as a footnote such as, note: percentages may not add up to 100 due to rounding. This helps the reader.

“This study explores the core topic and reflects strong analytical capacity, particularly for this large sample size and clear methods description. While the use of Univariate Logistic Regression to assess the association is appropriate for your objective, reporting of just p-values limits the interpretation of the findings. Presenting odds ratios with 95% CI would help enhance transparency and allow readers to better understand the magnitude and direction of association. Also, if the objective was only to check the association, then chi-square alone would have been sufficient. While I understand that everyone has their own way of doing the research study but it would be great if we do what’s specifically a statistical analyses are needed for, the authors have made methodologically validated their choice of univariable logistic regression, logistic regression is primarily intended for estimating the odds ratios and effect sizes, rather than solely for testing associations. For simple associations between categorical variables, Chi-square would have been sufficient. Using logistic regression without reporting the odds limits the study’s potential to yield a meaningful result. Nevertheless, the approach is understandable.”

Discussion

Although the discussion provides great details, it felt more like a thematic analysis than the standard discussion.

This is a very good and informative study.

Good luck!

Reviewer #2: Comments on “Factors associated with the contraceptive use among reproductive-age women during a pandemic: Evidence from a small developing state.

Dear Authors,

I would like to thank all the authors for conducting research and successfully turning it into a manuscript. Please find my reviews to help strengthen and support your study. These are my views and you have every right to disagree with it.

After reading it carefully, I noticed a discrepancy between methods and results. At first the authors stated that they have used logistic regression (LR) and then they have only reported the association with p-values. Which got me into thinking that if they only wanted to assess the association then why not use the Chi-square? Although the use of univariate logistic regression was mentioned and valid. But why complicate things that can be done in the simplest way rather than twisting and turning the purpose of standard statistical measures. For example, as a reader my simple understanding of the use of LR is to estimate odds and effect size and Chi-square for association. Even though you have clearly mentioned the reasons this does not follow a clear standard statistical measure. In context of validity, validity of a test refers to whether the test measures what it is supposed to measure? Isn’t it? So, if we take a look at it from that point of view? Did LR estimate the odds ratios? If yes, then where is the reporting of it?

Introduction sections should clarify Gender Based Violence in terms of SRH too

The discussion felt like more of a thematic analysis did authors create the theme? Or is it based on the sections that were used in the data collection tools? If so, then it would have been great if the authors clarified that in methods.

My only concern as a reviewer was the choice of statistical tests. Overall, the study is good, and it is very insightful.

Keep up the good work

Best Wishes,

Reviewer #3: According to the Data Availability Statement provided at the start of the manuscript PDF, the authors have not made all data underlying the findings fully available without restriction.

The statement explicitly specifies the following:

Availability: The data supporting the study's findings are available from the corresponding author rather than being in a public repository.

Condition: Access to the data is provided only "upon reasonable request".

While the journal (PLOS Global Public Health) generally requires authors to make all underlying data fully available without restriction before publication, the authors have indicated this restricted access model in their initial submission.

The authors explicitly state that the analysis was not intended to support causal inference or multivariable prediction. Key limitations include:

Lack of Multivariable Modeling: The analysis was restricted to univariable models. This means the study did not control for confounding variables; for example, the association between education and contraceptive use was not adjusted for age or relationship status.

Cross-Sectional Design: The design only captures a snapshot in time (April 28 to May 3, 2020), preventing any conclusions about changes in behavior over the course of the pandemic.

Data Quality Issues: Despite the 15% exclusion rule, the authors still analyzed alcohol and marijuana use because of their perceived importance, even though they exceeded the missing data threshold. They caution that these specific results should be interpreted as "for completeness only".

Exploratory Nature: The authors characterize the findings as exploratory and descriptive rather than definitive.

the manuscript is presented in an intelligible fashion and is written in standard English. The structure follows the traditional academic format for public health research, including a clear abstract, introduction, methods, results, and discussion section

Methodological considerations

Several limitations warrant careful consideration. First, the cross-sectional design precludes causal inference and does not allow assessment of temporal ordering between psychosocial distress and contraceptive behavior. Second, analyses were restricted to univariable models, limiting the ability to assess independent associations or account for confounding. Third, recruitment through cellphone text blasts and snowball dissemination may have resulted in overrepresentation of women with higher educational attainment or greater digital connectivity. As such, findings may not be fully generalizable to the broader population of reproductive-age women in Barbados.

Additionally, contraceptive use was self-reported and assessed over a two-month period without differentiation between consistent and inconsistent use. The analysis did not explicitly adjust for sexual activity or pregnancy intention, which may influence contraceptive behavior. Variables with substantial missing data, including alcohol and marijuana use, were interpreted cautiously and should not be overemphasized.

These limitations underscore that the findings should be interpreted as exploratory and hypothesis-generating rather than definitive.

Implications for policy and future research

Despite these constraints, the study provides context-specific insight into SRH patterns during the early phase of a global public health crisis in a small island developing state. The observed associations suggest that SRH policy in SIDS should consider the interplay between psychosocial stressors and reproductive health behavior during emergencies. Maintaining continuity of contraceptive services, expanding access to self-managed and long-acting methods where feasible, and integrating mental health screening within SRH services may strengthen resilience.

Future research should employ multivariable and longitudinal designs to disentangle pathways linking mental health, partnership dynamics, education, and contraceptive behavior. Studies that incorporate direct measures of sexual activity, fertility intentions, and service availability will be particularly important for clarifying unmet need and system-level vulnerabilities in SIDS contexts.

Despites all this, the manuscript aligns well with the mission of PLOS Global Public Health to highlight health inequities and provide evidence from underrepresented regions. Provided the data availability statement meets journal standards, it presents a compelling case for publication.

Reviewer #4: The manuscript addresses a significant public health issue in sexual and reproductive health and is methodologically sound, with clearly defined objectives, appropriate study design, and a coherent presentation of results.

The work provides valuable insights that contribute meaningfully to the field.

However, the manuscript does not clearly indicate compliance with journal requirements for deposition of research data in publicly accessible, community-recognized repositories. The authors are requested to confirm that all relevant raw and processed data have been deposited in appropriate domain-specific repositories and to provide the corresponding repository names, accession numbers, and a detailed Data Availability Statement to ensure transparency, reproducibility, and adherence to community standards.

6. PLOS authors have the option to publish the peer review history of their article (what does this mean?). If published, this will include your full peer review and any attached files.

**Do you want your identity to be public for this peer review?** For information about this choice, including consent withdrawal, please see our Privacy Policy.

Reviewer #1: No

Reviewer #2: No

Reviewer #3: **Yes:** Laetitia Berteline Bissaï

Reviewer #4: **Yes:** Joelma Baduro

Figure Resubmissions:

---

## [Editor Report · Decision Letter 1]

1 May 2026

Factors associated with contraceptive use among reproductive-age women during a pandemic: Evidence from a small developing state

PGPH-D-26-00542R1

Dear Dr. George,

We are pleased to inform you that your manuscript 'Factors associated with contraceptive use among reproductive-age women during a pandemic: Evidence from a small developing state' has been provisionally accepted for publication in PLOS Global Public Health.

Best regards,

Dr Tanmay Bagade, Ph.D., MS (O&G), MPH, MHM

Academic Editor